# ImageDriver: Let Vision-Language-Action Models Drive on 2D Images

## Abstract

Vision-language-action models (VLAs) in autonomous driving, which focus on 3D scene understanding and motion planning, confront a fundamental modality gap: pretrained only on image-text corpora, they inherently lack native 3D spatial comprehension. This limitation either yields coarse-grained textual interpretations of the driving scene or necessitates the integration of computationally expensive, auxiliary 3D modules. In this work, we challenge this prevailing convention by introducing ImageDriver, a novel VLA framework that circumvents the dependency on 3D data. It reformulates scene understanding and planning by recasting them as 2D object detection and 2D trajectory prediction tasks, executed directly on the image plane. By leveraging the intrinsic multimodal grounding of Vision-Language Models (VLMs), our method achieves a four-step pipeline: egocentric consistent perception, geometrically grounded reasoning, high-level meta-action prediction, and trajectory prediction, all in a fully differentiable and low-latency manner. We propose a two-stage knowledge-seeded policy optimization paradigm, which first fine-tunes ImageDriver on a multi-task mixed dataset to learn driving knowledge. To holistically optimize the agent's reasoning and decision-making, we further employ the Group Relative Policy Optimization (GRPO) algorithm to enforce end-to-end policy coherence across the complete VLA pipeline, from perception to planning. The superiority and versatility of our method are fully demonstrated by achieving state-of-the-art or competitive performance across detection, meta-action and trajectory prediction tasks.

## 1 Introduction

The emergence of Vision-Language Models (VLMs) (Achiam et al., 2024; Bai et al., 2025) has significantly advanced the end-to-end autonomous driving paradigm. In contrast to conventional methods (Hu et al., 2023; Jiang et al., 2023; Chen et al., 2024) that train perception and policy modules from scratch on driving data only, Vision-Language-Action Models (VLAs) (Jiang et al., 2025; Chi et al., 2025; Zheng et al., 2025), which build upon VLMs pretrained on web-scale data, integrate perception (vision), high-level reasoning (language), and decision-making (action and trajectory) abilities, thus promising superior generalization and a more nuanced understanding of complex scenarios.

However, the prevailing VLA paradigm confronts a fundamental modality gap. Current powerful VLMs are typically pretrained on vast corpora of images and text, which grounds their "understanding" firmly in the 2D image plane. Consequently, they lack the native 3D spatial comprehension that is conventionally considered essential for safe and precise vehicle control. To bridge this gap, current approaches have bifurcated into two suboptimal strategies, as shown in Fig. 1. The first relies on the VLA to generate coarse-grained textual interpretations of the driving scene (Jiang et al., 2025; Chi et al., 2025; Yuan et al., 2025), e.g., "There are many vehicles to my left", which lack the geometric precision required for accurate motion planning. The second, more common approach involves integrating auxiliary modules such as vision-based 3D object detectors or Bird's-Eye-View (BEV) converters to supply the requisite spatial information (Wang et al., 2025; Zheng et al., 2025). While functional, this integration is computationally expensive, increases system latency, and creates a complex, often brittle interface between the core language model and the specialized 3D perception components.

In this work, we challenge the prevailing convention that explicit 3D perception and prediction are prerequisites for end-to-end autonomous driving. We introduce ImageDriver, a VLA that circumvents the dependency on 3D data and 3D-aware models. Our key insight is that a feasible and safe 2D trajectory on the image plane should also be feasible and safe in the corresponding 3D space. ImageDriver employs a four-step pipeline that encompasses perception, reasoning, meta-action prediction, and planning. As shown in Figure 1, in the perception stage, our model takes its native multi-modal ability to ground its understanding by identifying and localizing all relevant traffic participants, *e.g.*, vehicles, pedestrians, and cyclists, as 2D bounding boxes directly on the egocentric consistent input image. These perceptual outputs are not merely coordinates; they form the factual basis for the subsequent reasoning stage. Here, the model leverages its vast pretrained knowledge to analyze the spatial relationships and implied dynamics of the detected objects, culminating in a high-level, interpretable meta-action. This action, such as "FORWARD, DECELERATE", represents the model's strategic driving intent. Finally, this strategic command guides the planning stage, which translates the abstract goal into a concrete and precise 2D trajectory on the image plane for the ego-vehicle to execute. This approach elegantly leverages the intrinsic strength of Vision-Language Models (VLMs) in multi-modal grounding, allowing the model to reason about and act upon the rich visual information it was originally trained on.

To facilitate this pipeline, we curated a dataset for **R**easoning with **b**ounding boxes, named nuScenes-RB-9k dataset from nuScenes (Caesar et al., 2020), a meticulously annotated collection featuring the geometrically-grounded planning rationales between 2D detection and high-level meta-actions, to train the model's reasoning capabilities explicitly. To holistically optimize the agent's behavior, we employ a two-stage knowledge-seeded policy optimization training paradigm, including supervised fine-tuning on a multi-task mixed dataset, and reinforcement learning using the Group Relative Policy Optimization (GRPO) (Shao et al., 2024b) algorithm. This training strategy enforces end-to-end policy coherence across the complete VLA pipeline, ensuring that all components, from perception to planning, are jointly optimized.

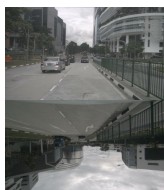

Figure 1: VLAs for autonomous driving. (a) Describing and reasoning with coarse-grained text, (b) Employing 3D tools or modules for perception and reasoning, (c) Our ImageDriver perceives, thinks, and drives on images.

To sum up, the key contributions of this work are as follows: (1) We propose a novel end-to-end VLA for autonomous driving, named ImageDriver, which uniquely reframes the entire driving task onto the 2D image plane. (2) We introduce "geometrically-grounded reasoning", which thinks with detected 2D bounding boxes and curated the nuScenes-RB-9k dataset to support it. (3) We employ a two-stage knowledge-seeded policy optimization training strategy combining supervised fine-tuning (SFT) with GRPO for knowledge acquisition and incentivization.

Extensive experimental results demonstrate that our ImageDriver is not a compromise but a powerful alternative. It achieves state-of-the-art or highly competitive performance across the distinct tasks of detection, meta-action prediction, and trajectory prediction. These results validate the superiority and versatility of our approach, presenting a more efficient, elegant, and computationally streamlined path toward building intelligent and capable autonomous driving systems.

## 2 RELATED WORK

### 2.1 VISION-LANGUAGE MODELS

The success of large language models (LLMs) (Yenduri et al., 2023; Brown et al., 2020; Touvron et al., 2023) has catalyzed the development of vision-language models (VLMs) (Radford et al., 2021; Zhu et al., 2023; Chu et al., 2023), which integrate visual and textual data for richer multimodal representations. Pioneering work like CLIP (Radford et al., 2021) aligns image and text features from separate encoders, enabling zero-shot prediction of correct image-text pairs. Building on these foundations, many contemporary VLMs—such as LLaVA (Liu et al., 2023)—conduct visual instruction tuning to acquire the multimodal instruction-following ability by learning a vision-language projector. wen2.5VL (Bai et al., 2025) and InternVL-3 (Zhu et al., 2025), in particular, employ native multimodal frameworks instead of CLIP to achieve superior multimodal understanding and grounding, enabling complex capabilities such as open-world object localization. More recently, DeepSeek-R1 (Guo et al., 2025) enhances the reasoning abilities of LLMs and VLMs by applying Group Reward Policy Optimization (GRPO) (Shao et al., 2024b) with simple, rule-based rewards. In this paper, we leverage the intrinsic multimodal grounding capabilities of VLMs, which are learned from web-scale data, for autonomous driving, and incentivize the reasoning ability of VLMs for more robust and safe driving.

### 2.2 AUTONOMOUS DRIVING

Autonomous driving has recently transitioned from traditional modular pipelines, i.e., perception, motion prediction, and planning, toward end-to-end learning-based paradigms (Hu et al., 2023; Jiang et al., 2023; Sun et al., 2024). UniAD (Hu et al., 2023) pioneered the integration of all sub-tasks into a cascaded framework, yielding substantial improvements over modular baselines. Subsequent works (Jiang et al., 2023; Ye et al., 2023; Chen et al., 2024) adopt bird's-eye view representations and generate planning trajectories through multi-stage interaction modeling. With the advent of vision-language models (VLMs), researchers have increasingly leveraged large language models (LLMs) and VLMs to enhance perception, reasoning, and decision-making. For instance, several approaches (Xu et al., 2024; Shao et al., 2024a) incorporate pretrained LLMs to produce driving actions accompanied by interpretable textual rationales. DriveVLM (Tian et al., 2024) introduces specialized reasoning modules for improved situational understanding, while DriveMM (Huang et al., 2024) processes multi-view video and image streams to enhance generalization in vehicle control. DriveMLM (Wang et al., 2023b) further extends this line by integrating a behavior-planning module that generates optimal driving decisions with explicit rationales. DriveMoE Yang et al. (2025), built on the embodied AI framework $\pi 0$ Black et al. (2024), introduces Action-MoE by training routing networks to dynamically activate expert modules for diverse driving behaviors. OmniDrive (Wang et al., 2025) replaced the CLIP visual encoder with a 3D visual encoder to generate object and map-related tokens, which are input to LLama (Touvron et al., 2023) for the final driving trajectory. OpenDriveVLA Zhou et al. (2025a) proposes an agent–environment–ego interaction paradigm for precise trajectory planning, while AutoVLA Zhou et al. (2025b) directly predicts semantic reasoning and trajectory plans from visual observations and language prompts. DriveAgent-R1 (Zheng et al., 2025), AutoDrive-R$^2$, and FutureSightDrive Zeng et al. (2025) employed GRPO and generate reasoning CoT to incentivize reasoning and self-reflection capacity for VLA.

## 3 METHOD

### 3.1 JUSTIFICATION

The central premise of our ImageDriver is that planning directly in the 2D image plane can serve as a valid and sufficient proxy for planning in 3D world space. This simplification is grounded in the geometric principles of perspective projection, which preserve critical properties related to trajectory feasibility and interaction safety. This section provides the theoretical justification for this approach.

**Trajectory Feasibility** For short-term motion planning, the road surface ahead of the ego-vehicle can be accurately approximated as a local plane. Based on that, we find that a 3D plane and its 2D image are related by a homography, i.e., a bijective projective transformation. This bijective mapping provides a strong theoretical guarantee: every point on the 3D drivable road plane corresponds

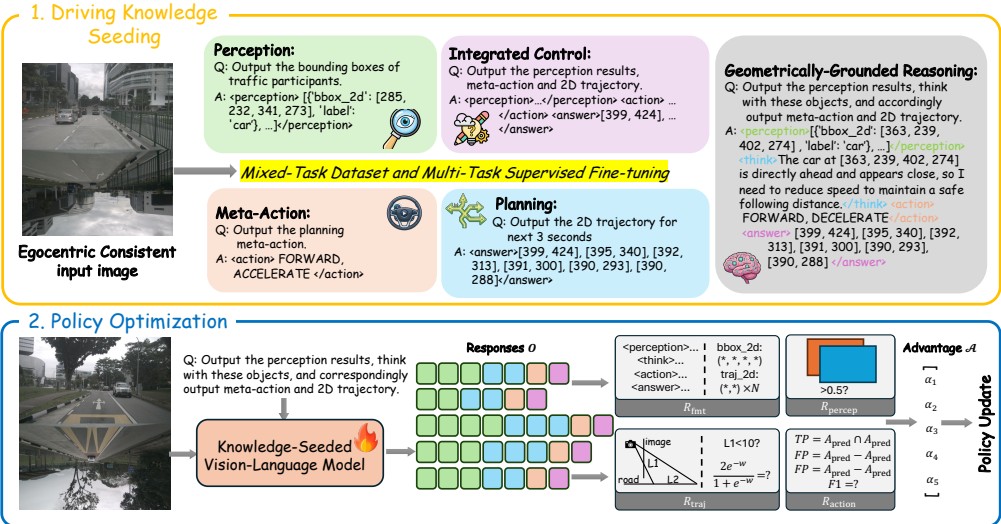

Figure 2: Training pipeline of ImageDriver. We propose a knowledge-seeded policy optimization training process. The first stage introduces nuScenes-RB-9k and uses a multi-task mixed dataset to seed driving knowledge into the model by SFT. The second stage utilizes GRPO on the complete VLA pipeline to holistically optimize the agent's behavior.

to a unique, predictable point on its 2D image projection, and vice versa. Therefore, a continuous and smooth trajectory planned on the 2D drivable surface is guaranteed to map to a continuous and smooth trajectory on the 3D road plane. While the real world contains non-planar surfaces (e.g., hills, banking), the local planarity assumption is a cornerstone of many validated approaches in autonomous driving and holds true for the vast majority of immediate planning scenarios. This ensures that a trajectory deemed feasible in 2D is also physically plausible in 3D.

**Interaction Safety** The safety of our paradigm is justified by an analogy to Configuration Space (C-Space) planning in robotics. We treat the 2D image plane as a simplified C-space, where the 2D bounding boxes of other vehicles act as Image-Space Obstacles (I-Space Obstacles). These I-Space Obstacles form a conservative superset of the actual projected 3D collision risks. This provides a strong safety guarantee: because a 3D collision must cause a 2D projection overlap, a trajectory that avoids all I-Space Obstacles is guaranteed to be collision-free. This approach also naturally handles occlusion. Since an I-Space Obstacle from occlusion is indistinguishable from one indicating a real collision threat, our model learns a conservative, risk-averse policy by avoiding all such regions.

## 3.2 KNOWLEDGE-SEEDED POLICY OPTIMIZATION

We present the Knowledge-Seeded Policy Optimization training paradigm of our ImageDriver for end-to-end autonomous driving. As illustrated in Fig. 2, we perform supervised fine-tuning to seed driving knowledge into the base model at stage 1 (Section 3.2.1) on a multi-task mixed dataset. Then, we employ GRPO on the complete VLA pipeline, including perception, reasoning, meta-action, and trajectory prediction to incentivize the agent's reasoning ability and holistically optimize its behavior.

### 3.2.1 STAGE 1: FOUNDATIONAL KNOWLEDGE SEEDING VIA SFT

The initial stage consists of supervised fine-tuning (SFT), a form of imitation learning designed to seed the VLA with a foundational driving policy. To this end, we train the model on a comprehensive, multi-task mixed dataset meticulously structured to address several key learning objectives:

**Perception** We use 2D projected object detection data to preserve the model's pre-trained visual grounding abilities and mitigate catastrophic forgetting. To generate 2D bounding box labels, we follow common practice (Wang et al., 2023a; Tang et al., 2024) and project the ground-truth 3D annotations from the nuScenes dataset onto the corresponding 2D camera image planes. Please refer to the Appendix A.2 for more details.

**Meta-Action Prediction** We use meta-action prediction data to develop an understanding of high-level driving intentions. Following the methodology of AlphaDrive (Jiang et al., 2025), we abstract the continuous ground-truth trajectories from the nuScenes dataset (Caesar et al., 2020) into a discrete set of high-level, interpretable meta-actions. Each meta-action $A$ comprises both a lateral and a longitudinal component, denoted as $A_{\mathrm{lat}}$ and $A_{\mathrm{lon}}$, which are derived based on the trajectory's terminal state. For lateral action derivation, we define three lateral commands: TURN LEFT, TURN RIGHT, and FORWARD. The determination is based on the final lateral displacement of the ego-vehicle's planned trajectory. Specifically, if the trajectory's terminal point has a lateral displacement exceeding $\tau_{\mathrm{lat}}$ to the left of the initial position, $A_{\mathrm{lat}}$ is labeled TURN LEFT. If the lateral displacement exceeds $\tau_{\mathrm{lat}}$ to the right, it is labeled TURN RIGHT. Otherwise, it is categorized as FORWARD. For longitudinal action $A_{\mathrm{lon}}$, we define four commands: ACCELERATE, DECELERATE, KEEP, and STOP. These are determined by the trajectory's final displacement and velocity. If the trajectory's terminal point is within a 0.1-meter longitudinal distance of the initial position and the final velocity is near zero, $A_{\mathrm{lon}}$ is classified as STOP. For non-stop trajectories, we compare the next-second velocity to the initial velocity. $A_{\mathrm{lon}}$ is labeled ACCELERATE or DECELERATE if the velocity increases or decreases, respectively, by more than a predefined threshold $\tau_{\mathrm{vel}}$. If the change in velocity is within this threshold, $A_{\mathrm{lon}}$ is labeled KEEP. We empirically set $\tau_{\mathrm{lat}} = 2.0$ m and $\tau_{\mathrm{vel}} = 0.1$ m/s.

**Planning** We use 2D trajectory data to hone precise, low-level motion planning skills. The 2D trajectory annotation is obtained using the same projection procedure as perception. Please refer to the Appendix A.2 for more details.

**Integrated Control** We employ three-step perception-action-planning sequences to foster multi-task learning and establish the causal link between observation and execution.

**Geometrically-Grounded Reasoning** In contrast to disciplines like mathematics and the formal sciences, which benefit from abundant, high-quality data for training reasoning capabilities, the nuanced decision-making process in autonomous driving is inherently difficult to capture. Furthermore, the manual annotation of such complex planning rationales is prohibitively expensive. Previous work (Wang et al., 2025; Chi et al., 2025; Jiang et al., 2025) has leveraged VLMs to generate Chain-of-Thought (CoT) data. However, these methods typically yield coarse-grained textual interpretations of the scene, resulting in ambiguous references to the objects in the image, while the geometric precision is essential for robust reasoning and motion planning. To address this deficiency, we introduce Geometrically-Grounded Reasoning and generate perception-planning rationales by distilling from advanced VLMs. Our approach prompts the Qwen-VL-Max model with a structured input comprising the ground-truth driving action $A$, the vehicle's ego-state, and the 2D bounding boxes $\mathcal{B}_{\mathrm{2D}}$ of surrounding agents. The model is tasked with generating a concise, causal reasoning process that logically connects the perceived environment to the given action. Finally, the generated rationales undergo a rigorous manual verification and filtering process, yielding a high-quality dataset consisting of 9k planning-centric reasoning data, named *nuScenes-RB-9k*. Please refer to the Appendix A.3 for more details. We use the four-step sequences from our nuScenes-RB-9k dataset to bootstrap the model's explicit reasoning faculties. Through the SFT stage, the model possesses a strong behavioral prior and can perform basic driving tasks in a manner consistent with the expert data.

### 3.2.2 STAGE 2: POLICY OPTIMIZATION VIA RL

Building upon the SFT-initialized model, this phase employs the Group Relative Policy Optimization (GRPO) (Shao et al., 2024b) algorithm to incentivize the VLA's higher-level reasoning and decision-making faculties. Different from conventional RL methods that depend on critic networks for value function approximation, GRPO introduces a mechanism based on the pairwise comparison of multiple candidate responses. This strategic shift away from an explicit value function not only simplifies the overall architecture but also reduces the computational demands of the training phase (Please see Appendix A.4). It commences by generating a set of $G$ distinct candidate responses, denoted as $O = \{o_1, \ldots, o_G\}$, for a given input prompt $q$ by sampling from the current policy. Specifically, we apply several rule-based reward functions to assess the response quality across the complete VLA output, from perception to planning:

**Format Reward** $R_{\mathrm{fmt}}$ is designed to enforce a strict and hierarchical output structure. It consists of three parts: (1) **Base structural reward** of 1.0 is granted if the generated responses in the form: `"<perception>2D bounding boxes here</perception> <think>thinking`

Table 1: Comparison on nuScenes object detection. [*] indicates sourced from (Tang et al., 2024). Best mAP within each category are in **bold**.

| Method | 2D/3D | mAP | Car | Truck | C.V. | Bus | Trailer | Barrier | Motor. | Bicycle | Ped. | T.C. |
|---|---|---|---|---|---|---|---|---|---|---|---|---|
| *VLM-Based 3D Driving Specialists* | | | | | | | | | | | | |
| OmniDrive-ViT | 3D | **40.7** | 57.5 | 36.4 | 15.9 | 40.3 | 16.5 | 47.2 | 41.3 | 45.7 | 59.0 | 57.3 |
| *Specialized 2D Object Detector* | | | | | | | | | | | | |
| StreamPETR[*] | 2D | 46.5 | - | - | - | - | - | - | - | - | - | - |
| MV2D[*] | 2D | 52.3 | - | - | - | - | - | - | - | - | - | - |
| DeformableDETR[*] | 2D | 50.2 | - | - | - | - | - | - | - | - | - | - |
| SimPB[*] | 2D | **54.1** | - | - | - | - | - | - | - | - | - | - |
| ImageDriver | 2D | **54.2** | 66.3 | 62.3 | 34.6 | 78.4 | 36.2 | 56.2 | 56.3 | 58.1 | 46.2 | 47.0 |

with bounding boxes here</think> <action>predicting meta-action here</action> <answer>2D trajectory here</answer>", otherwise 0. (2) **Perception format reward** ensures each bounding box in <perception> tags in in the format {"bbox_2d": [x_1, y_1, x_2, y_2]}. (3) **Trajectory format reward** of 1.0 is allocated for trajectory completeness and consistency, i.e., the planned path described within the ¡answer¿ tag must consist of exactly six trajectory points for the next 3 seconds at 0.5 interval.

**Perception Reward** $R_{\text{percep}}$ We follow previous work (Liu et al., 2025) and use Intersection over Union (IoU)-based reward. Specifically, a reward of 1 is granted if the IoU between the predicted and ground-truth bounding boxes exceeds a threshold of 0.5, and 0 otherwise. For multi-object, we use the Hungarian algorithm to find the matched bounding boxes and compute the mean IoU reward as the final perception reward.

**Meta-Action Reward** $R_{\text{action}}$ To evaluate the accuracy of the predicted high-level meta-action, we compute a reward based on the F1-score, which provides a harmonic mean of precision and recall between the predicted action and the ground-truth action set.

**Trajectory Reward** $R_{\text{traj}}$ We propose a L1 reward $R_{\text{traj-2D}}$ to prioritizes 2D trajectory adherence. Specifically, a reward of 1 is allocated if the L1 distance between the predicted and ground-truth 2D trajectory point is less than 10 pixels, 0 otherwise. Moreover, we re-project the 2D trajectory to the road plane in 3D space and calculate a sigmoid L2 reward as $R_{\text{traj-3D}} = \frac{2e^{-w}}{1+e^{-w}}$, where $w$ is the L2 distance between the re-projected predicted and ground-truth 3D trajectories.

The final reward is the weighted sum of the above reward terms:

$$R_{\text{acc}} = \lambda_{\text{fmt}} \cdot R_{\text{fmt}} + \lambda_{\text{percep}} \cdot R_{\text{percep}} + \lambda_{\text{action}} \cdot R_{\text{action}} + \lambda_{\text{traj}} \cdot R_{\text{traj}} . \tag{1}$$

## 3.3 EGOCENTRIC CONSISTENCY

Prevailing Vision-Language-Action (VLA) models often process a multi-view image collage, which typically arranges front and back camera feeds into separate rows. However, we contend that this composition introduces a significant *egocentric inconsistency*. For instance, a lane marking that appears as a left-turn arrow in the back-view image geometrically corresponds to a right-turn lane from the ego-vehicle's perspective, resulting in a counterintuitive and potentially misleading representation for robust perception and spatial reasoning. To address this issue, we propose a simple yet effective modification to the image collage construction: rotating the back view by 180 degrees prior to the vertical concatenation, as shown in Figure 1&2. The rationale for this transformation is twofold: (1) It corrects the egocentric inconsistency and establishes a coherent egocentric coordinate for accurate spatial reasoning and planning. (2) The vertical flip inherent in this rotation aligns the road surface visible at the bottom of the front view with the road surface at the top of the now-inverted back view, generating a visually continuous road plane across the entire image collage. As a result, the vehicle's path, both historical and projected, can be represented as a smooth and unbroken trajectory that flows seamlessly from the back view to the front view. This holistic representation simplifies the learning problem, providing the model with a more intuitive and contiguous basis for comprehending and predicting driving intentions.

Table 2: High-level meta-action prediction F1 score on the nuScenes dataset. † indicates trained on nuScenes. Best and second best results within each category are respectively in **bold** and underlined.

| Method | Lateral (F1) ↑ | | | Longitudinal (F1) ↑ | | | |
| --- | --- | --- | --- | --- | --- | --- | --- |
| | forward | left | right | keep | acc. | dec. | stop |
| Qwen2.5VL-7B | 64.67 | 24.15 | 30.85 | 40.73 | 55.14 | 51.41 | 41.82 |
| Qwen2.5VL-7B† | 94.46 | 63.00 | 67.01 | 57.62 | 74.35 | 77.10 | 75.00 |
| ImageDriver | **96.82** | **75.51** | **75.71** | **61.23** | **81.76** | **80.19** | **81.80** |

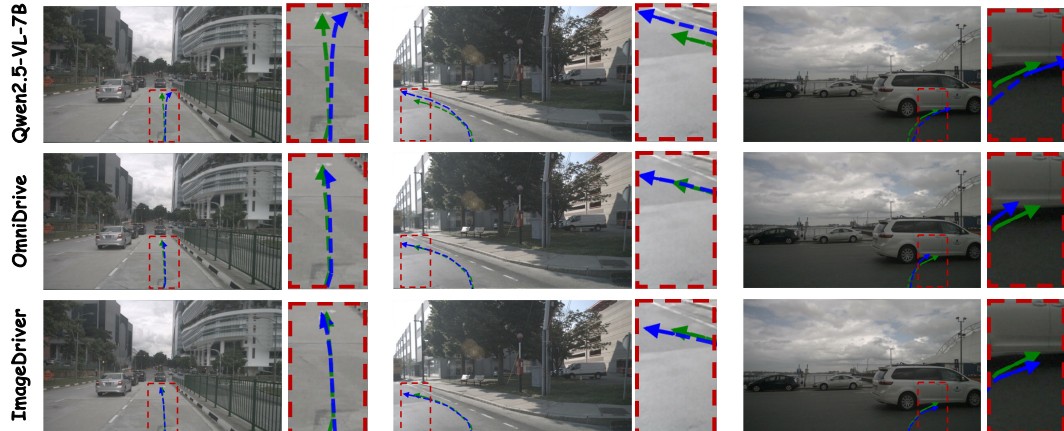

Figure 3: Visualization of 2D trajectories across Qwen2.5-VL-7B, OmniDrive, and our ImageDriver on the nuScenes validation dataset. The predicted and ground-truth trajectories are depicted in blue and green, respectively.

## 4 EXPERIMENTS

### 4.1 IMPLEMENTATION AND METRICS

We use Qwen2.5-VL-3B and 7B, a powerful open-source VLM, as the base model for ImageDriveer. Training and inference are conducted on 8 A800 GPUs. The maximum pixels is configured as 720,000. For SFT, i.e., stage 1, we fine-tune the model for 2 epochs on the mixed dataset in a multi-task manner to seed the model with knowledge about perception, reasoning, high-level meta-action making, and 2D trajectory prediction. During Stage 2, reinforcement learning, we fine-tune the model trained by SFT using GRPO to incentivize the reasoning ability of the trained policy. The number of completions, i.e., $G$ is set to 8. For perception evaluation, we report the mAP for 2D object detection. Since no confidence is assigned to the bounding boxes $\hat{\mathcal{B}}_{2D}$ output by VLMs, we set all the confidences to 1. For meta-action prediction, we use the F1-score for all lateral and longitudinal meta-action classes. For planning evaluation, we employ the L2 distance (in meters) between the predicted and ground-truth trajectories, and we report the displacement error at future horizons of 1s, 2s, and 3s, along with the average error. Additionally, following BEV-Planner (Li et al., 2024), the Collision Rate and Intersection Rate with the road boundary are adopted to evaluate the safety of the planning.

### 4.2 VISUALIZATION

Figure 3 presents a qualitative comparison of our method against other approaches on the nuScenes dataset. Notably, Qwen2.5-VL-7B struggles to generate accurate predictions, exhibiting significant trajectory deviations, particularly in turning scenarios. Although OmniDrive demonstrates better performance, its trajectories are often overly aggressive, with predicted speeds substantially exceeding the ground truth. In contrast, our method, ImageDriver, consistently generates reliable and conservative plans that closely align with the ground truth, as explained in Section 3.1.

## 4.3 MAIN RESULTS

**Perception** We first evaluate the perception capability of ImageDriver in Tab. 1. Notably, few VLM-based autonomous driving methods report object detection metrics, with OmniDrive being the only exception. OmniDrive replaces the original CLIP (Radford et al., 2021) backbone in LLava Liu et al. (2023) with the 3D vision encoder of SteamPETR (Wang et al., 2023a), which is trained via a detection proxy task. However, its mAP is substantially lower than that of SteamPETR (62.0), likely due to multi-task interference. In contrast, our VLA model achieves state-of-the-art performance, even compared to specialized 2D object detectors trained on nuScenes (Wang et al., 2023a;c; Zhu et al.; Tang et al., 2024), despite not being explicitly optimized for 2D object detection.

**Meta-Action** The performance of ImageDriver on meta-action prediction is detailed in Table 2. Our model demonstrates superior performance across all lateral (Path) and longitudinal (Speed) action categories when evaluated by the F1-score. Compared to the base Qwen2.5VL-7B model, our Supervised Fine-Tuning stage (Qwen2.5VL-7B†) provides a dramatic performance uplift, particularly for turning maneuvers where the 'left' F1 score improves from 24.15 to 63.00. Building on this strong foundation, ImageDriver achieves the highest scores in every category, such as 96.82 for 'forward', 75.51 for 'left' turns, and 81.76 for 'acceleration'. This comprehensive superiority underscores the effectiveness of our proposed training methodology.

**Trajectory Prediction** In the open-loop trajectory prediction task, as detailed in Table 3, ImageDriver achieves a competitive average L2 error of 0.40m. While this significantly surpasses general-purpose VLMs, it does not reach the state-of-the-art precision of specialized models like AutoDrive-$R^2$ (0.19m). This discrepancy is an anticipated consequence of our Driving on Image paradigm, which directly predicts trajectories as pixel coordinates in the 2D image. The precision of this approach is inherently constrained by the coarse

Table 3: Open-loop trajectory prediction L2 errors (m) on the nuScenes dataset. (where [1], [2] and [3] indicate sourced from (Qiao et al., 2025), (Xing et al., 2025) and (Hwang et al., 2024)). Best results within each category are in **bold**.

| Method | L2 Error (m) ↓ | | | |
|---|---|---|---|---|
| | 1s | 2s | 3s | **Avg.** |
| *Open-source Generalist VLMs* | | | | |
| LLaVA-1.6-Mistral-7B[2] | 1.49 | 3.38 | 4.09 | 2.98 |
| Llama-3.2-11B-Vision-Instruct[2] | 1.54 | 3.31 | 3.91 | 2.92 |
| Qwen2-VL-7B-Instruct[2] | 1.45 | 3.21 | 3.76 | 2.81 |
| DeepSeek-VL2-16B[1] | 0.66 | 1.68 | 2.92 | 1.75 |
| DeepSeek-VL2-28B[1] | **0.37** | 1.35 | 2.96 | 1.56 |
| LLaMA-3.2-11B-Vision-Instruct[1] | 0.52 | 1.42 | 2.68 | 1.54 |
| LLaMA-3.2-90B-Vision-Instruct[1] | 0.66 | 1.71 | 3.01 | 1.79 |
| Qwen-2.5-VL-7B-Instruct[1] | 0.46 | **1.33** | **2.55** | **1.45** |
| *Training-based Driving Specialists* | | | | |
| UniAD[3] | 0.42 | 0.64 | 0.91 | 0.66 |
| VAD[3] | 0.17 | 0.34 | 0.60 | 0.37 |
| BEV-Planner[3] | 0.16 | **0.32** | **0.57** | **0.35** |
| Ego-MLP[3]* | **0.15** | **0.32** | 0.59 | **0.35** |
| *Ours and Key Competitors (Specialized Driving VLAs)* | | | | |
| DriveVLM[3] | 0.18 | 0.34 | 0.68 | 0.40 |
| OmniDrive[3] | 0.14 | 0.29 | 0.55 | 0.33 |
| DriveVLM-Dual[3] | 0.15 | 0.29 | 0.48 | 0.31 |
| EMMA (random init)[3] | 0.15 | 0.33 | 0.63 | 0.37 |
| EMMA[3] | 0.14 | 0.29 | 0.54 | 0.32 |
| EMMA+[3] | 0.13 | 0.27 | 0.48 | 0.29 |
| Imprompt-VLA | **0.13** | 0.27 | 0.53 | 0.30 |
| AutoDrive-$R^2$ 7B | **0.13** | **0.19** | **0.25** | **0.19** |
| ImageDriver | 0.17 | 0.36 | 0.66 | 0.40 |

feature maps produced by computationally efficient large vision models, which often have high downsampling ratios (e.g., 28x). This quantization introduces a lower bound on the achievable accuracy when the 2D pixel predictions are back-projected into the 3D world. Thus, our model's performance represents a trade-off between the benefits of a holistic, image-based reasoning system and the precision limits imposed by the underlying vision encoder's resolution. The safety and feasibility of the planned trajectories are evaluated in Table 5, where ImageDriver consistently achieves state-of-the-art or best-in-class performance. For the Collision Rate, ImageDriver records the lowest average error of all methods at 0.26%, matching the best specialist models. This underscores its superior ability to maintain safe distances from other agents. For Intersection Rate, which measures trajectory feasibility with respect to the drivable area, ImageDriver again shows strong results. It obtains the best average rate (1.77%) among its VLA-based peers. This comprehensive performance in safety-critical metrics validates the effectiveness of our model's decision-making process.

## 4.4 ABLATION STUDY

**Knowledge-Seeded Policy Optimization** To validate our two-stage Knowledge-Seeded Policy Optimization (KSPO) strategy, we conducted an ablation study (Table 4) by training variants with only Supervised Fine-Tuning (SFT) or Reinforcement Learning (RL). We find that while both methods provide performance gains, the SFT-only variant surpasses the RL-only model. This suggests that RL, on its own, is inefficient at navigating the vast search space of our task's structured, multi-step reasoning process (perception → reasoning → meta-action → planning). SFT is therefore essential for "seeding" the model with a coherent policy and a foundational understanding of the required causal chain. The superior performance of the complete ImageDriver model, which combines both stages, confirms that our hybrid approach is critical: SFT provides the necessary knowledge foundation, which RL then effectively refines to achieve optimal results.

**Supervised Fine-Tuning** During the SFT stage, we train the base model, Qwen2.5-VL-7B, on a mixed dataset, which includes data from 2D object detection, reasoning with bounding boxes, meta-action prediction, and trajectory prediction. Moreover, to ensure the model's correct understanding and reasoning, the training images are concatenated in an ego-centric consistent way. The ablation study further investigates the contributions of our mixed dataset and multi-task training, as well as the effect of ego-consistent image input. Excluding the mixed-task training dataset and training on the proposed nuScenes-RB-9k ('w/o. Mixed Data') results in a performance degradation, increasing the average L2 error to 0.43m. More significantly, removing our proposed egocentric consistent input ('w/o. Ego. Cons.') leads to a

Table 4: Ablation studies of trajectory planning L2 errors on the nuScenes dataset to validate each proposed component.

| Method | L2 Error (m) ↓ | | | |
|---|---|---|---|---|
| | 1s | 2s | 3s | Avg. |
| Qwen2.5-VL-7B | 0.52 | 1.46 | 3.78 | 1.92 |
| Qwen2.5-VL-7B + SFT | 0.21 | 0.41 | 0.76 | 0.46 |
| Qwen2.5-VL-7B + RL | 0.23 | 0.43 | 0.82 | 0.49 |
| w/o. Mixed Data | 0.18 | 0.39 | 0.72 | 0.43 |
| w/o. Ego. Cons. | 0.20 | 0.45 | 0.84 | 0.50 |
| w/o. $R_{\text{traj-2D}}$ | 0.17 | 0.38 | 0.70 | 0.42 |
| w/o. $R_{\text{traj-3D}}$ | 0.18 | 0.40 | 0.75 | 0.44 |
| w/o. $R_{\text{percep}}$ | 0.17 | 0.37 | 0.69 | 0.41 |
| w/o. $R_{\text{action}}$ | 0.18 | 0.39 | 0.68 | 0.40 |
| ImageDriver | 0.17 | 0.36 | 0.66 | 0.40 |

substantial drop in accuracy, with the error rising to 0.50m. We owe this to the inconsistent ego-camera representation and discontinuous trajectory. This finding highlights the critical role of a geometrically consistent input for precise trajectory planning.

**Reinforcement Learning** We dissect the contributions of each component within our composite reward function used during the Reinforcement Learning (RL) stage. As shown in Table 5, individually ablating trajectory reward $R_{\text{traj}}$ (including $R_{\text{traj-2D}}$ and $R_{\text{traj-3D}}$), IoU-based perception reward $R_{\text{percep}}$ and action reward, and each leads to a discernible increase in the average L2 error, rising to 0.42m, 0.44m, 0.41m, and 0.41m respectively. This confirms that these components all positively contribute to the final planning accuracy. And the most important Reward is $R_{\text{traj-3D}}$. We believe this is because 3D L2-based $R_{\text{traj-3D}}$ make a great alignment with the evlautaion metric, i.e., L2 error. Among the reward components, the 3D trajectory reward, $R_{\text{traj-3D}}$, proves to be the most impactful. We attribute its significance to the direct alignment between its formulation, which is based on 3D L2 distance, and the final evaluation metric of L2 error.

## 5 CONCLUSION

In this work, we introduce ImageDriver, a novel VLA that challenges the reliance on computationally expensive 3D data in autonomous driving. By reformulating scene understanding and planning as 2D tasks executed directly on the image plane, our model circumvents the modality gap inherent in many VLAs. This is enabled by the Knowledge-Seeded Policy Optimization paradigm that uses SFT to seed foundational knowledge, then RL to refine strategic reasoning. Our experiments demonstrate the efficacy of this approach, with ImageDriver achieving state-of-the-art or competitive performance across perception, meta-action prediction, and planning. While our approach excels in safety and high-level reasoning, we acknowledge a trade-off in its fine-grained trajectory precision, which is constrained by the vision encoder's resolution. Future work will focus on mitigating this bottleneck and extending the framework to more complex, long-horizon scenarios.

# 6 ETHICS STATEMENT

This research adheres to the ethical guidelines of the ICLR community. Our work focuses on developing decision-making and planning methods for autonomous vehicles and does not involve the collection of new, sensitive personal information or data that may compromise individual privacy.

All data used in this study is derived from the nuScenes dataset, a publicly available benchmark that has been released under an appropriate license for research purposes. The dataset creators have already taken steps to anonymize data, such as blurring faces and license plates. Our custom-curated nuScenes-DoI-9k dataset consists only of new annotations and rationales overlaid on this existing public data. We have carefully ensured full compliance with the dataset's usage policies.

Potential societal impacts of our work are twofold. On the positive side, our method may advance the state-of-the-art in autonomous driving, potentially improving road safety, transportation efficiency, and accessibility. On the negative side, as with any autonomous agent research, there exists the risk of model failure leading to accidents, as well as the potential for misuse of the underlying technology in surveillance or military applications. We acknowledge these risks and emphasize that our work is intended solely for academic research and beneficial civilian applications.

No new human subjects, personally identifiable information (PII), or harmful synthetic content were involved in this study. We believe the ethical risks of this work are minimal and have been appropriately managed.

# 7 REPRODUCIBILITY STATEMENT

We are committed to ensuring the full reproducibility of our results, in accordance with established machine learning research guidelines.

- Code and Data Release: We will release our core implementation code upon publication. Crucially, we will also release the full annotation files for our nuScenes-DoI-9k dataset, along with the scripts used for data processing and generation, to allow the community to build upon our work.

- Datasets: The base dataset used in our experiments, nuScenes, is publicly available and can be accessed from its official source.

- Hyperparameters: We provide complete details of our ImageDriver's hyperparameters (including learning rates, batch sizes, optimizers, training epochs, and reward function weights for both the SFT and RL stages) in the Appendix.

- Architecture and Model Details: Detailed descriptions of our ImageDriver architecture, which is based on the Qwen2.5-VL-7B model, are reported in the Method section.

- Computational Environment: All experiments were conducted on NVIDIA A800 GPUs. We report key computational statistics, including model size, in the experimental section to facilitate comparison.

We believe these measures are sufficient for independent researchers to reproduce and verify our results fully.

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

## A  APPENDIX

### A.1  USE OF LARGE LANGUAGE MODELS (LLMS)

In the preparation of this manuscript, we utilized a large language model (LLM), specifically Google's Gemini, as a writing assistant. The model was employed to aid in refining, polishing, and improving the clarity and academic tone of the text based on the authors' directives and content. The core scientific contributions—including the initial concepts, experimental design, implementation, and the final analysis and interpretation of results—are exclusively the work of the human authors. The LLM was not used to generate novel scientific insights, formulate hypotheses, or conduct experiments. All text and suggestions provided by the LLM were critically reviewed, edited, and verified by the authors to ensure they accurately represent our own work and findings. The ultimate responsibility for the scientific integrity, correctness, and all claims made in this paper rests entirely with the authors.

### A.2  DETAILS ABOUT DATASET CURATION

**2D annotation generation** This process leverages the provided sensor calibration and vehicle pose data to transform 3D coordinates from the global frame to the 2D pixel frame for each camera. For each point $\mathbf{P}_{\text{lidar}} = (x, y, z)$ in 3D bounding box corners set $\mathcal{C}_{\text{lidar}}$ or trajectory waypoint set $\mathcal{T}_{\text{lidar}}$ (including history and future trajectory) in the LiDAR coordinate system, we augment it to $\tilde{\mathbf{P}}_{\text{lidar}} = (x, y, z, 1)$, which is then transformed to a image point $\mathbf{P}_{\text{img}} = (u, v)$ using:

$$\mathbf{P}_{\text{img}} = (u, v) = (\frac{u'}{d'}, \frac{v'}{d'})$$

$$\mathbf{P'}_{\text{img}} = (u', v', d') = \mathbf{K} \cdot \mathbf{T}_{\text{cam}\leftarrow\text{ego}_\text{c}} \cdot \mathbf{T}_{\text{ego}_\text{c}\leftarrow\text{ego}_\text{l}} \cdot \mathbf{T}_{\text{ego}_\text{l}\leftarrow\text{lidar}} \cdot \tilde{\mathbf{P}}_{\text{lidar}},$$

(2)

with transforms: $\mathbf{T}_{\text{ego}_\text{l}\leftarrow\text{lidar}}$ from LiDAR to the ego frame of LiDAR, $\mathbf{T}_{\text{ego}_\text{c}\leftarrow\text{ego}_\text{l}}$ from ego frame of LiDAR to that of camera, $\mathbf{K}$ from camera to image plane. $d'$ is the depth in the image coordinate system. Therefore, we can obtain the projected 2D corners $\mathcal{C}_{\text{2D}}$ of the 3D box corners and the $\mathcal{T}_{\text{2D}}$.

Owing to perspective distortion, the projected vertices $\mathcal{C}_{\text{2D}}$ do not typically form an axis-aligned rectangle. The final 2D bounding box is therefore derived by first computing the convex hull of the projected 2D points. This polygonal hull is subsequently clipped against the image canvas boundaries and formulated as $\mathcal{H} = \{\mathbf{H} = (u, v)\}^8$. The axis-aligned 2D bounding box, denoted $\mathcal{B}_{\text{2D}}$, is then defined by the extrema of the resulting vertices in $\mathcal{H}$, i.e.,

$$\mathcal{B}_{\text{2D}} = (\min(u), \min(v), \max(u), \max(v)).$$

(3)

Moreover, we observe that the 3D waypoints that are physically proximate to the ego-vehicle and have depth $d'$ near 0 cannot be validly projected onto the image plane due to the perspective division by depth in Eq. 2. To maintain a complete trajectory representation, we represent these unprojectable waypoints in the final 2D trajectory $\mathcal{T}_{\text{2D}}$ using dedicated special tokens.

In this way, we obtain the 2D bounding boxes $\mathcal{B}_{\text{2D}}$ and trajectory points $\mathcal{T}_{\text{2D}}$. Note that the history 3D waypoints are typically projected on the back view, and the future points are on the front view.

### A.3  PROMPTS TO GENERATE GEOMETRICALLY-GROUNDED PLANNING RATIONALES

The prompt to generate geometrically-grounded planning rationales is given in Figure 5

### A.4  GRPO ALGORITHM FOR RL

To quantify the relative quality of all responses given the rewards $\{R_1, \ldots, R_G\}$, GRPO normalizes these rewards by subtracting the group mean and dividing by the standard deviation. then, the advantage for each response can be calculated as:

$$\mathcal{A}_i = \frac{R_i - \text{mean}(\{R_i\}_{i=1}^G)}{\text{std}(\{R_i\}_{i=1}^G)},$$

(4)

where $\mathcal{A}_i$ is the relative advantage of the $i$-th answer. Then a regularization term is incorporated in the optimization objective function to ensure the updated policy $\pi_\theta$ remains close to the old reference policy $\pi_{\mathrm{ref}}$. This is achieved by adding a KL-divergence term $D_{\mathrm{KL}}(\cdot \,\|\, \cdot)$ to the loss function:

$$J_{\mathrm{GRPO}}(\theta) = \mathbb{E}_{q \sim P(Q), \{o_i\}_{i=1}^{N} \sim \pi_{\theta_{\mathrm{old}}}(O|q)}$$

$$\left[ \sum_{i=1}^{G} \frac{\pi_\theta(o_i \mid q)}{\pi_{\theta_{\mathrm{old}}}(o_i \mid q)} \cdot \mathcal{A}_i - \beta D_{\mathrm{KL}}(\pi_\theta \,\|\, \pi_{\mathrm{ref}}) \right], \tag{5}$$

where $\beta$ acts as a hyperparameter to balance the trade-off between exploration and old policy during optimization.

### A.5 MORE IMPLEMENTATION DETAILS

Our training methodology consists of two distinct stages: Supervised Fine-Tuning (SFT) followed by Reinforcement Learning (RL).

**Stage 1: Supervised Fine-Tuning (SFT).** The model is fine-tuned for 2 epochs using a comprehensive, mixed-task dataset. We employ the AdamW optimizer with a peak learning rate of $5.0 \times 10^{-6}$ and a cosine decay schedule. To accommodate large batch sizes, we use a per-device batch size of 2 with 8 gradient accumulation steps, resulting in an effective batch size of 16 per device. Input images are processed to a maximum resolution of 720,000 pixels, and training is conducted with bfloat16 mixed-precision.

**Stage 2: Reinforcement Learning (RL).** Building on the SFT checkpoint, the model is further optimized for 1 epoch using the Group Relative Policy Optimization (GRPO) algorithm. The policy is updated with a learning rate of $1 \times 10^{-6}$ and a global batch size of 16. To regularize the policy update and prevent catastrophic forgetting of the SFT-learned behaviors, we apply a KL-divergence penalty with a coefficient of $1 \times 10^{-2}$. During training, we sample 8 responses per prompt to estimate the policy gradient. The reward weight is all set to 1.

### A.6 VISUALIZATION OF PERCEPTION

The visualization of 2D object detection of our ImageDriver is shown in Figure 4.

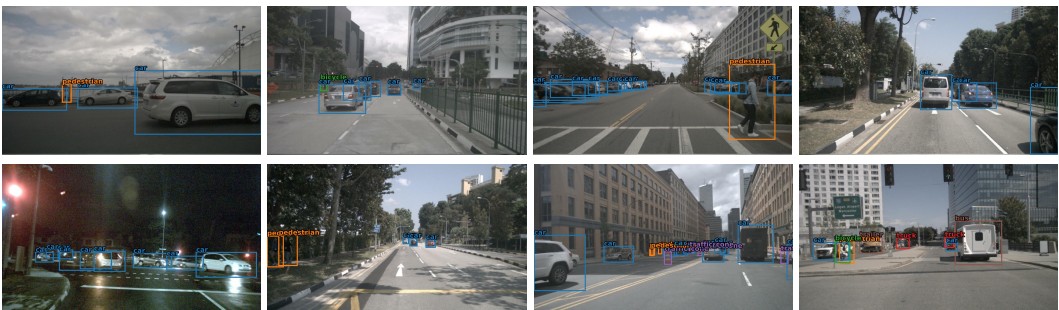

Figure 4: Visualization of 2D object detection of our ImageDriver on the nuScenes validation dataset.

### A.7 COLLISION AND INTERSECTION

The collision rate with other driving agents and the intersection rate with the boundary of the drivable surface are given in Table 5.

Table 5: Collision rate and intersection rate with the road of trajectory prediction on the nuScenes dataset. Best results are in **bold**.

| Method | Collision (%) ↓ | | | | Intersection (%) ↓ | | | |
|---|---|---|---|---|---|---|---|---|
| | 1s | 2s | 3s | Avg. | 1s | 2s | 3s | Avg. |
| *Training-based Driving Specialists* | | | | | | | | |
| UniAD | 0.02 | 0.25 | 0.84 | 0.37 | **0.20** | **1.33** | **3.24** | **1.59** |
| VAD | 0.04 | 0.27 | 0.67 | 0.33 | 0.21 | 2.13 | 5.06 | 2.47 |
| Ego-MLP | **0.00** | 0.27 | 0.85 | 0.37 | 0.27 | 2.52 | 6.60 | 2.93 |
| BEV-Planner | **0.00** | 0.29 | 0.73 | 0.34 | 0.35 | 2.62 | 6.51 | 3.16 |
| *Training-based Driving Specialists* | | | | | | | | |
| OmniDrive | **0.00** | 0.13 | 0.78 | 0.30 | 0.56 | 2.48 | 5.96 | 3.00 |
| ImageDriver | **0.00** | **0.11** | **0.66** | **0.26** | 0.50 | 1.58 | **3.24** | 1.77 |

PROMPT_FORMAT = """
You are the decision-making AI for an autonomous vehicle. You are analyzing a composite image created by stitching a front-facing camera view with a vertically-flipped rear-facing camera view.

**Current Speed:** {speed} m/s

**The traffic participants in the front view have been detected:**
{perception}

**Your Task:**
Your determined driving decision is to "{action}". Given the current speed and the list of participants, identify only the key objects in the front view whose presence and position directly force this decision. For each selected object, provide a brief explanation of its impact.

**Key Spatial Context:**
To accurately judge the position of other vehicles, use the following reference points and rules. In the standard bounding box format [x1, y1, x2, y2], the x-coordinates (x1, x2) represent the horizontal position on the image.
1. The top half of the image (area where y < 448) represents the FRONT VIEW. Objects here are in front of your vehicle.
2. The bottom half of the image (area where y > 448) represents the REAR VIEW. Objects here are behind your vehicle.
3. The center of your current lane, directly in front of your vehicle, corresponds to the horizontal coordinate x=392.
4. Calculate an object's horizontal center using (x1 + x2) / 2.
5. If an object's horizontal center is near 392, it is likely in your direct path.
6. If its horizontal center is significantly lower than 392, it is to your left.
7. If its horizontal center is significantly higher than 392, it is to your right.

**Important constraints:**
1. Present your analysis as a real-time thought process.
2. Please strictly DO NOT include "{action}" in your response to avoid confusion and leakage. For example, do not say, "Based on the instruction to decelerate, I conclude..."
3. Keep the explanation concise.
"""

Figure 5: Prompt to generate geometrically-grounded planning rationales.

