# OpenReview forum: "ImageDriver: Let Vision-Language-Action Models Drive on 2D Images"
_ICLR.cc/2026/Conference — ICLR 2026 Conference Withdrawn Submission_

### Official Review · Reviewer_p4nH · 2025-10-24

**Soundness:** 3
**Presentation:** 3
**Contribution:** 2
**Rating:** 4
**Confidence:** 5

**Summary:**

This paper introduces ImageDriver, a vision-language-action (VLA) framework for autonomous driving that performs scene understanding and trajectory planning purely in the 2D image plane, without relying on explicit 3D information. The method builds on pretrained Vision-Language Models (VLMs) and employs a two-stage knowledge-seeded policy optimization strategy combining supervised fine-tuning and GRPO-based reinforcement learning. The authors also curate a new nuScenes-RB-9k dataset to support geometrically grounded reasoning with 2D bounding boxes. The experiments demonstrate competitive results in detection, meta-action prediction, and trajectory planning compared to both 2D and 3D driving models.

**Strengths:**

1. The two-stage policy optimization approach, combining supervised and reinforcement learning, is experimentally validated through detailed ablations.
2. The curated nuScenes-RB-9k dataset contributes valuable data resources to support grounded reasoning and explainable VLA-based driving.

**Weaknesses:**

1. Although the paper asserts that every point on the 3D drivable road plane corresponds to a unique point on its 2D image projection and vice versa, this only holds under a known homography. In practice, mapping from 2D to 3D requires depth information, which is not available to the VLM. Consequently, without 3D box awareness or depth cues, the approach has limited applicability to real-world autonomous driving systems where depth reasoning is critical.
2. The model jointly outputs perception, meta-action, planning, and reasoning modules, but the real-time feasibility of such an end-to-end pipeline is questionable. The paper does not provide latency analysis or runtime comparisons, which are essential for on-road deployment.
3. The paper claims that GRPO incentivizes the VLA’s higher-level reasoning and decision-making faculties. Yet, it lacks a clear explanation or discussion of why the long reasoning is important for autonomous driving.
4. In Table 1, the comparison between 2D and 3D detection results is conceptually inconsistent. Since ImageDriver operates without camera intrinsics/extrinsics, achieving valid depth estimation or 3D detection is infeasible, which undermines the significance of this comparison.
5. Figure 3 shows visual comparisons where both Qwen2.5-VL and OmniDrive produce acceptable driving behaviors. In several examples (e.g., in the last column, the front car is running), all models avoid collisions despite imperfect alignment with ground truth.
6. According to prior works such as BEV-Planner, open-loop evaluation on nuScenes does not adequately reflect closed-loop driving performance. The paper would benefit from a closed-loop simulation to substantiate its claims.

**Questions:**

N/A

---

### Official Review · Reviewer_Jnsj · 2025-10-29

**Soundness:** 2
**Presentation:** 3
**Contribution:** 2
**Rating:** 2
**Confidence:** 5

**Summary:**

This paper proposes to use 2d image plane for trajectory planning instead of depending on 3D data. Their key insight is that the ego-centric image and the road outline represent the driving scene sufficiently and a trajectory planned on the 2D drivable surface is enough to maintain a smooth trajectory in 3D. This would reduce a lot of computation either from calculating 3D projection like Birds eye view or utilizing multiple sensors like lidar. They curated a dataset of 2D bounding boxes and rules of creating trajectories using them from NuScenes dataset. They used a two stage training process with SFT and GRPO similar in nature to the AlphaDrive paper to train a VLA based planner. Their results show that 2D approximation does not worsen performance massively from using 3D methods.

**Strengths:**

1. Efficiency. Calculation of 3D co-ordinates or creation of BEV or multi-sensor fusion are all computationally expensive steps. If 2D image plane can safely work, it reduces a lot of extra compute.
2. Geometrically grounded reasoning. This approach of stitching together front and rear images to represent the driving scene in 2D is novel and can be efficient scene representation.

**Weaknesses:**

1. Performance hit. Even though the authors claim their approach is not a compromise, comparison presented in table 3 shows that state of the art methods are all significantly better, especially in longer horizon planning task. This is not surprising, but there should be some discussion on how to address this.
2. Lack of generalization. The work is shown only using NuScenes dataset and the prompt for geometric grounding is clearly customized for NuScenes. It is not clear if this can generalize to other datasets.
3. Reproducibility. The manual curation of the dataset for SFT makes this work hard to reproduce.

**Questions:**

1. Are the front and rear image being used only for creation of the ego-centric image? This process is not explained clearly, as planning trajectory in image plane could mean just using the front image.
2. The authors claim efficiency, however there is no mention of FPS or any latency metrics. They should compare against other SOTA models on latency, number parameters etc to make this claim.

---

### Official Review · Reviewer_CQ7S · 2025-10-31

**Soundness:** 3
**Presentation:** 3
**Contribution:** 3
**Rating:** 4
**Confidence:** 3

**Summary:**

This paper proposes ImageDriver, a novel Vision-Language-Action (VLA) framework for end-to-end autonomous driving purely on 2D images, without relying on 3D or BEV representations. It reformulates driving as a sequence of 2D perception, reasoning, meta-action, and trajectory prediction tasks, trained through a two-stage Knowledge-Seeded Policy Optimization (KSPO) combining supervised fine-tuning and Group Relative Policy Optimization (GRPO). The approach achieves competitive detection and planning results, supported by the new nuScenes-RB-9k dataset.

**Strengths:**

1. Novel paradigm: Recasts autonomous driving fully in 2D image space, offering a simple yet effective alternative to 3D BEV-based frameworks.
2. Integrated reasoning pipeline: Connects perception, reasoning, and planning in one differentiable structure, improving coherence and latency.
3. KSPO training: Combines multi-task SFT with GRPO reinforcement, effectively aligning perception and decision-making stages.
4. Dataset contribution: The proposed nuScenes-RB-9k fills a gap in reasoning-grounded 2D driving data.
5. Practical engineering insight: Egocentric back-view rotation and stitching are simple yet significantly improve spatial continuity.

**Weaknesses:**

1. Evaluation inconsistency: Mixed 2D and 3D mAP results hinder fair comparison; should be clearly separated.
2. Limited robustness analysis: Only front/back egocentric fusion tested; missing side-view and adverse-weather evaluations.
3. Incomplete 2D→3D analysis: No systematic study of input resolution or upsampling vs. trajectory accuracy trade-offs.
4. Dataset characterization: Lacks analysis of RB-9k coverage, distribution, and cross-domain generalization.
5. Missing latency/cost details: No latency, FLOPs, or memory usage reported despite claims of low computational overhead.
6. RL training metrics: Absence of convergence curves or stability discussion.
7. Performance gap: Still lags behind specialized 3D-VLA systems in absolute planning precision.

**Questions:**

1. How robust is egocentric fusion to lateral and occluded views?
2. How is 2D–3D back-projection accuracy affected by input resolution?
3. What is the end-to-end latency and compute cost compared to 3D-BEV methods?

---

### Official Review · Reviewer_5TQP · 2025-11-01

**Soundness:** 2
**Presentation:** 2
**Contribution:** 2
**Rating:** 4
**Confidence:** 4

**Summary:**

This paper proposes a vision-language-action (VLA) model framework named ImageDriver, which aims to address the "modality gap" problem in existing autonomous driving VLA models. The core idea of ImageDriver is to challenge the mainstream paradigm that "3D perception is a prerequisite for autonomous driving" by completely reconstructing the entire driving task—from scene understanding to motion planning—onto the 2D image plane.

**Strengths:**

1、Novel Core Argument: The paper's most outstanding merit lies in its bold core idea—performing end-to-end driving directly on 2D images.  This directly challenges the prevailing paradigm in the autonomous driving field, which is centered on Bird's-Eye View (BEV) and reliant on explicit 3D representations.

2、Rigorous Dataset Construction: The construction of the nuScenes-RB-9k dataset is commendable.  The authors not only use 2D bounding boxes as perceptual input but also employ structured prompts to have Qwen-VL-Max generate chains of reasoning directly associated with these bounding boxes.  These chains are then subjected to manual verification and filtering.

3、Ego-centric Consistency: Furthermore, the proposed "ego-centric consistency" input processing—rotating the rear-view camera image 180 degrees—is a simple yet ingenious technique.  It geometrically unifies the coordinate systems of the front and rear views, thereby providing the model with a more intuitive representation for learning continuous trajectories.

**Weaknesses:**

1、Issues with the New Paradigm: Proposing a new paradigm is commendable. However, several questions remain. For instance, the details of how the paradigm maps 2D trajectories back to the 3D physical space are unclear. Without precise depth information, the accuracy of this back-projection is questionable. Furthermore, the paradigm is not intuitively convincing. Autonomous driving demands accurate 3D understanding, and an approach that circumvents this requirement may inherently be limited to sub-optimal solutions.

2、Dependence on Camera Intrinsics and Extrinsics: The model predicts trajectories directly in the pixel space, which implies its output is highly coupled with the specific camera setup (intrinsic and extrinsic parameters) used during training. This raises the question: if the model were deployed on a new vehicle with different camera mounting positions or models, would it require complete retraining?

3、Scalability to Multi-Camera Systems: The proposed "ego-centric consistency" stitching method adeptly handles the front and rear cameras. However, modern autonomous driving systems typically employ surround-view configurations with six or even more cameras. How does the framework scale to handle these more complex systems? Simple image stitching for surround-view scenarios often introduces significant distortions, which would severely compromise the intuitiveness and effectiveness of planning on a 2D plane.

4、Lack of Real-Time Performance Metrics: While the authors claim the method is computationally efficient, the paper lacks crucial real-time performance metrics, such as inference latency and frames per second (FPS). These metrics are critical for autonomous driving systems, as decisions with latency beyond a certain threshold can lead to significant safety risks.

5、Inconsistent Dataset Naming: There is an inconsistency in the dataset's naming. It is referred to as "nuScenes-RB-9k" in the Abstract and Introduction sections, but as "nuScenes-DoI-9k" in the Ethics Statement section. The authors should unify the name for consistency.

**Questions:**

Refer to Weaknesses.

---

### Note · Authors · 2025-11-12

I have read and agree with the venue's withdrawal policy on behalf of myself and my co-authors.